# Etiopathogenetic Factors of Hepatocellular Carcinoma, Overall Survival, and Their Evolution over Time—Czech Tertiary Center Overview

**DOI:** 10.3390/medicina58081099

**Published:** 2022-08-14

**Authors:** Petr Hříbek, Johana Klasová, Tomáš Tůma, Tomáš Kupsa, Petr Urbánek

**Affiliations:** 1Department of Internal Medicine, 1st Faculty of Medicine, Charles University and Military University Hospital Prague, 169 02 Prague, Czech Republic; 2Department of Internal Medicine, Faculty of Military Health Sciences, University of Defense, 500 02 Hradec Kralove, Czech Republic; 3Department of Radiodiagnostic, Military University Hospital Prague, 169 02 Prague, Czech Republic

**Keywords:** hepatocellular carcinoma, liver cirrhosis, transarterial chemoembolization, BCLC classification

## Abstract

*Background and Objectives*: Hepatocellular carcinoma (HCC) is the most common form of primary liver cancer with a highly unfavorable prognosis. *Aims*: Retrospective statistical analysis of patients with HCC in the field of liver cirrhosis treated at our center from the perspective of demography, and the effects of key changes in diagnostic and therapeutic procedures in the last 10 years on overall survival (OS) and earlier diagnosis. *Materials and Methods*: This study included 170 cirrhotic patients with HCC (136 men, 80%). Demographic and etiological factors and OS were analyzed based on distribution into three groups according to the period and key changes in diagnostic and therapeutic approaches (BCLC classification staging; standardization of protocol for transarterial chemoembolization (TACE) and the introduction of direct-acting antivirals (DAA) for the treatment of chronic viral hepatitis C (HCV); expansion of systemic oncological therapy). *Results*: The mean age at the time of diagnosis was 69.3 years (SD = 8.1), and etiology was as follows: non-alcoholic steatohepatitis (NASH) 39%, alcoholic liver disease (ALD) 36%, HCV 18%, cryptogenic liver cirrhosis 3%, chronic hepatitis B infection (HBV) 2%, and other etiology 2%. Distribution of stages according to the BCLC: 0 + A 36%, B 31%, C 22%, and D 11%. However, the distribution in the first studied period was as follows: 0 + A 15%, B 34%, C 36%, and D 15%; and in the last period: 0 + A 45%, B 27%, C 17%, and D 11%, and difference was statistically significant (*p* < 0.05). The median OS for stages 0 + A, B, C, and D was 58, 19, 6, and 2 months, respectively. During the monitored period, there was a visible increase in the etiology of ALD from 30% to 47% and a decrease in HCV from 22% to 11%. In patients treated with TACE (stage B), the median OS grew from 10 to 24 months (*p* < 0.0001) between the marginal monitored periods. *Conclusions*: We described a decreasing number of patients with HCV-related HCC during follow-up possibly linked with the introduction of DAA. In our cohort, an improvement in early-stage diagnosis was found, which we mainly concluded as a result of proper ultrasound surveillance, the institution of a HCV treatment center, and increased experience of our sonographers with an examination of cirrhotic patients. Lastly, we described significantly improved overall survival in patients with intermediate HCC treated by TACE, due to the increased experience of interventional radiologists with the method at our facility and an earlier switch to systemic therapy in case of non-response to TACE.

## 1. Introduction

Liver cirrhosis is a stereotypical result of many chronic liver diseases if not diagnosed on time and adequately treated. Among the most common causes of death in patients with liver cirrhosis is HCC. It is a primary malignant liver tumor, which typically develops mostly in the field of liver cirrhosis. HCC is a malignancy with a very unfavorable prognosis, whereby the median survival of untreated patients in advanced stages is 3.4 months [1]. On the contrary to this, early diagnosed patients who undergo radical surgical therapy, including liver transplantation, have a 5-year survival rate of 65–78% [2]. Despite the progress made in the systemic therapy of HCC, overall survival in late stages has changed only a little. The overall unfavorable prognosis is caused by the late diagnosis of the disease, with only one-third of patients being diagnosed in the early stages of the disease and having the possibility of radical and potentially curative therapy [3]. Because patients with liver cirrhosis of any etiology, patients with HBV, and patients with NASH, are at high risk of developing HCC, monitoring is recommended. This monitoring consists of a liver ultrasound examination at 6-month intervals [4]. When diagnosing a suspected lesion at liver ultrasound in a risk patient, it is necessary after establishing of certain diagnosis of HCC to determine the stage of the disease. For these purposes, dynamic contrast-enhanced computed tomography and/or magnetic resonance are used. The most used staging system for HCC is, concerning the almost regular presence of liver cirrhosis, different from other solid tumors that used TNM staging. Thus, in addition to tumor characterization, liver function is evaluated as well. This staging is called the Barcelona classification according to the Barcelona Clinic Liver Cancer (BCLC) and directly determines the relevant therapy method [3]. HCC has been among the crucial hepatogastroenterology issues at the Department of Medicine 1st Faculty of Medicine Charles University and Military University Hospital Prague (MUH) for more than 10 years. In this synoptic article, the authors systematically describe retrospective collected demographic data of a comprehensive cohort of patients who were monitored and treated in the department from 2011 to 2021. The impact of certain changes in the treatment procedures, as applied over time and which represented significant changes according to the authors, are documented.

## 2. Materials and Methods

The cohort included all patients with diagnosed HCC in the field of liver cirrhosis between 1 January 2011 and 31 December 2020 who were monitored and treated at the Department of Internal Medicine at the 1st Faculty of Medicine of Charles University and the MUH.

The diagnosis of HCC was based on valid international guidelines [4,5,6] and the national guidelines of the Czech Society of Hepatology [7] and was determined in patients with liver cirrhosis according to non-invasive diagnostic criteria on imaging techniques (multiphase computed tomography, dynamic contrast magnetic resonance) and/or histology. The Institutional Review Board was consulted and concluded that there is no need for a letter of acceptance due to the purely retrospective nature of this study.

The patient cohort was divided into 3 groups according to the date of the HCC diagnosis (periods 1, 2, and 3). The division was performed concerning the key changes in the treatment allocation strategy and the introduction of standardization and/or new therapeutic modalities for HCC. The periods were structured as follows in Table 1:

The hypothesis states that the partial changes in the care of HCC patients, as presented by the division into individual periods, will have an effect on prolonging overall survival, changing the composition of patients in terms of etiology, and increasing the frequency of diagnosis of the disease in its early stages.

The authors assessed the frequency of different etiologies of liver cirrhosis for each subgroup and different stages of HCC according to the BCLC in the analyzed period. In addition, we performed a Kaplan–Meier analysis of overall survival over the entire monitored period and the three aforementioned periods, as well as compared survival in the individual stages in the respective periods.

The survival curves were compared using the log-rank test, whereby a *p*-value <0.05 was determined to be statistically significant. The testing of empirical distribution was used to determine statistical significance between BCLC stages in the studied periods. The median follow-up was determined based on the Kaplan–Meier reverse analysis. The statistical assessment was performed using MedCalc^®^ software, version 20.106 (MedCalc Software Ltd., Ostend, Belgium).

## 3. Results


Entire monitored period 2011–2021:


The characteristics of the study population are provided in Table 2. With respect to the above-mentioned criteria, the cohort overall included 170 patients with HCC (136 men, 80% of the cohort), of which the mean age at the time of diagnosis was 69.3 years (SD = 8.1). The etiology of liver cirrhosis in these patients during the entire monitored period included NASH in 39%, ALD in 36%, HCV in 18%, cryptogenic liver cirrhosis in 3%, HBV in 2%, and other etiologies—hereditary hemochromatosis, coinfection HBV + chronic hepatitis D (HDV) in 2%. The distribution of the stages according to the Barcelona classification in the years 2011–2021 was as follows: 0 + A 36%, B 31%, C 22%, and D 11%. The median follow-up was 68 months (95% Cl 45–82). The median overall survival (OS) for stages 0 + A, B, C and D were 58 (95% Cl 35–76), 19 (95% CI 15–22), 6 (95% CI 4–10) and 2 months (95% CI 1–5), respectively (Figure 1). The differences in OS between the individual stages were statistically significant (*p* < 0.05).
Time period 1 (2011–2013):

A total of 53 patients (42 men, 79%) were diagnosed during these years, of which the mean age at the time of diagnosis was 68.1 years (SD = 8.3). The share of the etiologies was as follows: NASH 38%, ALD 30%, HCV 22%, HBV, and cryptogenic cirrhosis both 4%, and hereditary hemochromatosis 2%. The share according to the BCLC stages were 0 + A 15% (among these patients underwent 13% liver transplantation, 62% liver resection, and 25% ablative therapy), B 35% (among these patients underwent 89% TACE, 11% best supportive care), C 35% (among these patients 61% were treated by sorafenib, 37% best supportive care (BSC)), and D 15%, with a median OS of 76 months (95% CI 17–106), 10 months (95% CI 5–18), 5 months (95% CI 1–7), and 2 months (95% CI 0–8), while the differences in OS between the stages were statistically significant (*p* < 0.05). Overall survival rates in 1, 3, and 5 years are shown in Table 3.
Time period 2 (2014–2017):

A total of 64 patients with HCC (78% men) were diagnosed in our facility during this time, of which the mean age was 69.1 years (SD = 8.2). In terms of etiology, NASH was the most prevalent at 44%, followed by ALD at 33%, HCV at 19%, HBV at 3%, and cryptogenic cirrhosis at 1%. The shares according to the BCLC stages were 0 + A 45% (among these patients underwent 25% liver transplantation, 25% liver resection, 25% ablative therapy, 25% TACE), B 33% (among these patients underwent 71% TACE, 19% sorafenib treatment, 10% BSC), C 14% (among these patients 40% were treated by sorafenib, 60% BSC), and D 8%, with a median OS of 52 months (95% CI 35–72), 21 months (95% CI 13–34), 6 months (95% CI 1–14) and 2 months (95% CI 1–4), respectively. The differences in OS between the stages were statistically significant (*p* < 0.05). Overall survival rates in 1, 3, and 5 years are shown in Table 4.
Time period 3 (2018–2021):

During the last monitored period, HCC was diagnosed in 53 patients (83% men), of which the mean age at the time of diagnosis was 70.7 years (SD = 7.6). As for the etiology, ALD was present in 47%, NASH in 36%, HCV in 11%, cryptogenic cirrhosis in 4%, and HBV + HDV coinfection in 2%. The share according to the BCLC stages was 0 + A 45% (among these patients underwent 24% liver transplantation, 28% liver resection, 20% ablative therapy, 28% TACE), B 27% (among these patients underwent 86% TACE, 14% BSC), C 17% (as a first-line therapy was among these patients in 45% used sorafenib, 22% atezolizumab + bevacizumab, 22% lenvatinib and 11% BSC), and D 11%. The median OS for stage 0 + A was not reached (mean of 37 months, SD = 3.05), for stage B 24 months (95% CI 13–27), C 10 months (95% CI 1–18), and D 3 months (95% CI 1–5). The differences in OS between the stages were statistically significant (*p* < 0.05). Overall survival rates in 1, 3, and 5 years are shown in Table 5.
Comparison of the share of etiologies of HCC in the individual time periods (shown in Figure 2):

The three most common etiologies of HCC were ALD, NASH, and HCV, which always accounted for more than 90% of all the HCC in the monitored time periods. The other etiological factors for developing HCC were marginal. During time periods 1, 2, and 3, the share of ALD as an etiology of HCC grew from 30% to 33%, and subsequently to 47%, respectively. The share of NASH as an etiology of HCC in time periods 1, 2, and 3 varied and accounted for 38%, 44%, and 36%, respectively. The percentage for the whole group was 39%. The share of HCV as an etiology of HCC in the particular time periods was 22%, 19%, and 11%, respectively.
Comparison of the share of the stages according to the BCLC classification in the individual time periods (shown in Figure 3):

The frequency of stage 0 + A according to the BCLC classification was in the individual time periods 15%, 45%, and 45%, respectively. For stage B, this was 34%, 33%, and 27%, respectively. For advanced stage C, it was 36%, 14%, and 17%, respectively. Finally, for the terminal HCC stage D, it was 15%, 8%, and 11%, respectively. The difference between stages was evaluated by testing empirical distribution and was statistically significant with *p* < 0.05.
Comparison of overall survival by stages in the individual time periods (shown in Figure 4):

The stages are marked with a letter, as indicated in the BCLC classification, and the number of the time period (e.g., 0 + A1 for BCLC stage 0 + A in the period 2011–2013).
○Stage 0 + A: Despite the seemingly dramatic difference, the differences between median OS in groups 0 + A1 (76 months), 0 + A2 (52 months), and 0 + A3 (47 months) were not statistically significant (*p* = 0.51).○Stage B: The differences between the median OS for groups B1, B2, and B3 (10, 21, and 24 months, respectively) were statistically significant (*p* < 0.05).○Stage C: The differences between the median OS for groups C1, C2, and C3 (5, 6, and 10 months, respectively) were not statistically significant (*p* = 0.61).○Stage D: The differences between the median OS for groups D1, D2, and D3 (2, 2, and 3 months, respectively) were not statistically significant (*p* = 0.33).

## 4. Discussion

This paper summarizes the long-term experience of one center for therapy of HCC and analyses the impact of key changes in the therapeutic approach to this disease and HCV disease, which is a significant risk factor for HCC.

The group included only patients with liver cirrhosis to maintain the homogeneity of the cohort. The diagnostic and treatment procedure in these patients followed the algorithm of the Barcelona classification; the approach to the non-cirrhosis patients is different and cannot be compared [8]. Based on the experience of the authors, the percentage of patients at the center with HCC in a non-cirrhotic field is increasing over time. A publication by Polish authors offers a degree of comparative material concerning a demographically similar population, but includes patients without liver cirrhosis [9] too.

Our cohort was unequivocally dominated by men (80% of patients with HCC), which is a slightly higher percentage compared to the large population setups [10] and the aforementioned work from Poland [9]. Our data correspond with the oncological reports in the Czech Republic, where the incidence for the year 2018 in men was 5.74/100,000 persons/year and in women was 2.26/100,000 persons/year [11].

The mean age of 69.3 years (SD = 8.1) at the time of the diagnosis corresponded, in general, with the known fact that it is a disease of older age and that this did not significantly change over time. From the first view, this fact could probably play an important role in the overall unfavorable prognosis of HCC. This is due to many patients seeming unable to undergo radical treatment due to their overall condition and comorbidities, even though cancer itself might not be so advanced. However, according to an extensive analysis comparing the treatment effects in elderly patients with HCC, this has not been confirmed, with older patients benefitting from the entire range of available therapeutic modalities [12].

In this study, the most common etiological factor of cirrhosis complicated by the development of HCC was NASH (39%), followed by ALD (36%) and HCV (18%). Other etiological factors were rare and variable. Although the annual risk of developing HCC in NASH-associated cirrhosis is lower compared to ALD and HCV (e.g., according to Ascha et al., 2.6% of the patients [13]), the prevalence and hence the importance of NASH is growing concerning the HCC issue [14,15]. NASH in our cohort oscillated around an average of 39% (accounting for 38%, 44%, and 36% in the individual time periods, respectively). During the monitored period, there was a visible and quite surprising growth in ALD (30% vs. 47%, 16 vs. 25 patients in absolute numbers). The authors do not have a clear explanation for this phenomenon. A higher rate of examinations performed by general practitioners could be the reason, although this is speculation. In addition, the amounts of alcohol intake discovered from medical histories do not always unequivocally correspond with the generally established risk of developing liver cirrhosis. In this case, the effect of genetics, which is not routinely examined, could be the reason, although this is also pure speculation. For instance, the SERPINA1 gene with genotypes MZ and MS, in this case, represents a protective factor for the development of fibrosis [16], whereas, for example, the polymorphism of the PNPLA3 rs738409 C > G gene is a risk factor for both liver fibrosis and the development of HCC [17].

There was a significant decrease in HCV as an etiological factor of HCC (22% vs. 11%, 12 vs. 6 patients in absolute numbers). This trend is connected to the change in the number and profile of patients with HCV infection in recent years. The number of patients with a newly diagnosed HCV associated with cirrhosis has decreased dramatically. The majority of previously diagnosed patients with liver cirrhosis associated with HCV have already been successfully cured of HCV infection—they achieved sustained virologic response (SVR) by using DAA. The potential benefit of achieving SVR after DAA treatment from the perspective of fibrosis regression probably occurs with some latency and is interfered with by several negative factors (presence of metabolic syndrome, alcohol intake) [18]. For this reason, the effect of DAA therapy could not have been visible in the second studied period, however, is possible it positively affected the last studied period. The reason for the decrease in the number of patients with HCV-related HCC is probably decreasing in the number of patients with HCV-related advanced fibrosis as the earlier stages of HCV-related fibrosis are not considered the risk factor for HCC. In the conditions of our facility, the next possible factor was the large pool of HCV patients who failed in treatment with interferon-based regimes before DAA. The strategy after introducing DAA was the treatment of the most advanced patients at that time with many of them already with HCC before the start of therapy. Additionally, only then treatment of next patients according to liver stiffness from highest to patients without significant fibrosis. So, in the first studied period, there were an abnormal number of patients with advanced fibrosis and cirrhosis related to HCV which led to a higher incidence of HCC. So, there probably were two main factors for decreasing HCV-related HCC over time: firstly, a large number of untreated cirrhotic patients with HCV at the baseline of this study, and secondly there was a positive effect of DAA therapy due to fibrosis reversal after achieving SVR.

The data showing the real representation of the individual stages according to the BCLC are not consistent, they are often a secondary output of studies focused on specific types of treatment or etiology. An older publication by Llovet et al. [19] offers an estimate based on randomized controlled trials (RCT), where stage 0 + A accounted for 30%, B + C 50%, and terminal stage D for 20% of the cases. During the entire monitored period, the ratio in our cohort was very similar: stage 0 + A 36%, B 31%, C 22%, and D 11%. However, the difference was interesting and statistically significant, with the individual stages in the first monitored period being as follows: 0 + A 15%, B 34%, C 36%, and D 15%; and in the last time period 0 + A 45%, B 27%, C 17%, and D 11%. The authors attribute this shift to these factors. The first is the new possibility of HCV treatment, as discussed above, whereby many referred patients may not have been previously monitored. Additionally, in part of them, early HCC was detected while they were waiting for treatment. The crucial change was that before 2011, there was no instituted systemic ultrasound surveillance for all risk groups of patients. The next reason is, understandably so, training and super specialization of the sonographers responsible for the monitoring of the risk patients at the center. Nowadays, HCC surveillance in patients at risk is the most common indication for ultrasound examinations in our department. Additionally, the growing awareness of HCC among the professional public is among the reasons for earlier referral of HCC patients from other hospitals to our center. Further, the increasing numbers of patients with cirrhosis of all etiologies encountered in our care from other health providers led to a larger pool of patients undergoing HCC surveillance in our center.

By analyzing survival in the individual BCLC stages during the entire monitored period, the median OS for stages 0 + A, B, C, and D were found to be 58, 19, 6, and 2 months, respectively. The estimated average median OS in many RCTs [20,21,22] was approximately 30 months for treated patients of stage B [23]. However, these are highly selected groups of patients, and it would seem that the outcome of the meta-analysis by Lencioni et al., 19.4 months, is closer to data from real practice [24]. Within this context, the statistically significant prolongation of the median OS of patients with intermediate HCC between the marginal monitored periods in our cohort, 10 and 24 months (*p* < 0.05), is interesting. In 2014, the introduction of a standardized protocol for indicating and performing TACE in our center, the unification of the therapy response evaluation using the Modified Response Evaluation Criteria in Solid Tumor (mRECIST) [25] system, as well as improving the experience of interventional radiologists with locoregional therapy of these patients, could all have contributed to the better results. We did not observe a statistically significant change in overall survival in any of the other stages. As surgical resection is the standard therapy of treatment for many other malignant diseases of the liver, the experience of surgeons with this kind of surgery at our facility was high before the first studied period of this study, and outcomes were consistent. The same reason explains consistent outcomes for transplantation, which is realized only in the two centers in the Czech Republic with which we closely collaborate. Until the end of 2018, sorafenib was the only standard systemic treatment modality for HCC in the Czech Republic. From that year, atezolizumab + bevacizumab and lenvatinib were first-line therapy modalities in addition to sorafenib, but only for a limited number of patients. Even so, sorafenib was still the most used systemic therapy. This is related to the last determined time period (2018–2021), where, despite the expansion of the systemic therapy for the advanced stage of HCC, a statistically significant difference in the median OS was not achieved. The reason for this could be the short length of the monitored period, whereby the effects of the improved therapy options did not fully show. However, the authors consider the main limitation to be the small size of the cohort of patients (*n* = 37, with 9 patients in the time period 2018–2021) and the fact that nearly half of the patients were still treated by sorafenib.

## 5. Conclusions

The authors summarize aspects of 10 years of experience and the effects of changes in therapy for hepatocellular carcinoma and HCV infection on a cohort of patients from one center. A decrease in HCC patients with etiological HCV infection was demonstrated. On the contrary, a surprising increase in ALD patients as a cause for liver cirrhosis complicated by HCC presented itself. This trend contrasts with the generally accepted increase in the incidence of HCC associated with NASH-related cirrhosis. Adequate ultrasound surveillance has a considerable impact on the early diagnosis of HCC. In the group, a 3-fold increase in the detection of the early stages of HCC was recorded. This can be attributed to the increased experience of sonographers, greater awareness of related expertise, the necessity to monitor risk groups, and the targeted search for HCV patients. In the reported group of patients, a statistically significant impact on the prolongation of the median OS was observed in the intermediate stage of HCC. This was attributed to the increasing experience of interventional radiologists with locoregional therapy, more precise indications for TACE, and an earlier switch to systemic therapy when there is an unsatisfactory response to local destruction methods.

## Figures and Tables

**Figure 1 medicina-58-01099-f001:**
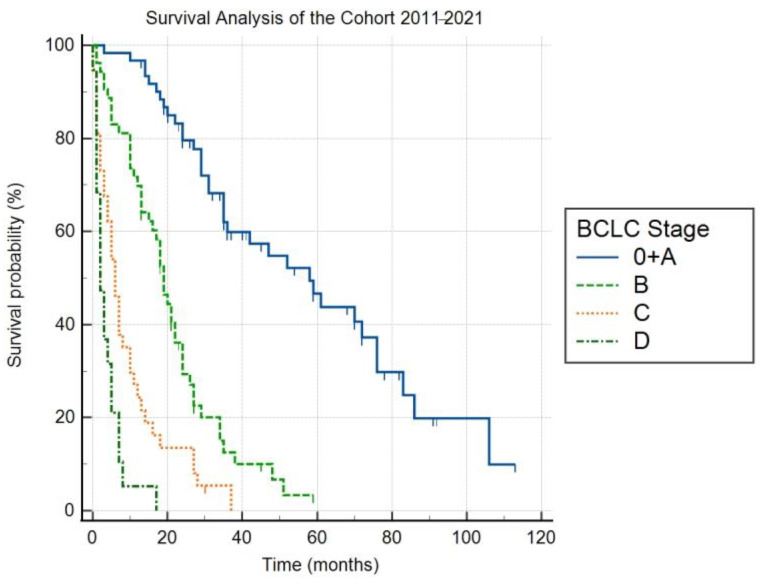
Survival Analysis of the Cohort 2011–2021.

**Figure 2 medicina-58-01099-f002:**
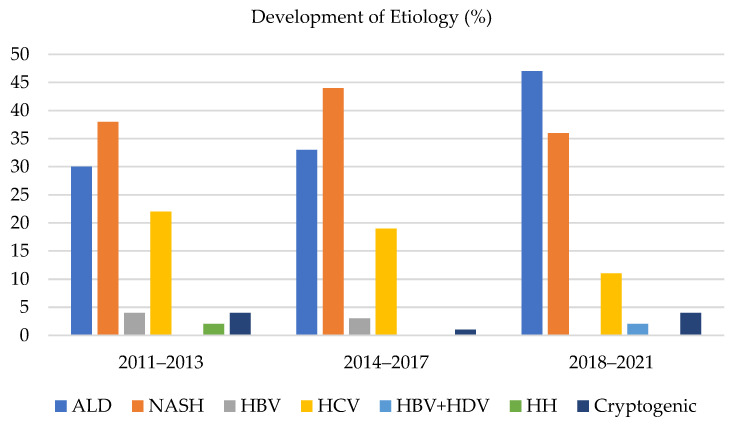
Overview of etiologies in % according to the studied time periods. A continuously increasing proportion of ALD-related HCC in the cohort and a nearly linear decrease in HCV as the etiological factor of HCC are shown.

**Figure 3 medicina-58-01099-f003:**
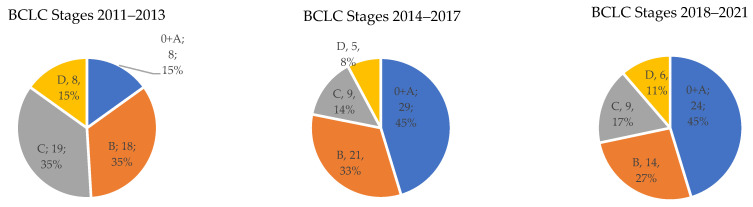
Comparison of BCLC stages according to the studied time periods. It is shown that initial changes in approach to patients with risk of developing HCC (systematic application of HCC surveillance) and classification according to the BCLC classification led to the change in the distribution of the initial stage at the time of diagnosis to earlier stages. This effect was sustained to the end of the studied period.

**Figure 4 medicina-58-01099-f004:**
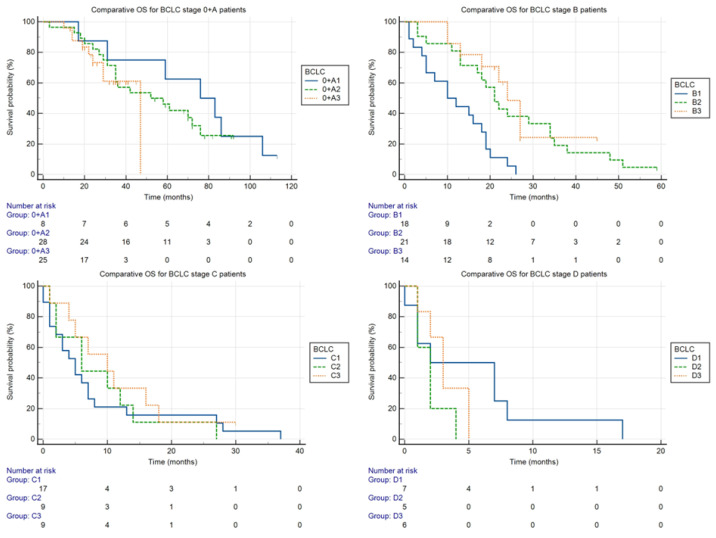
Comparison of BCLC stages according to the studied time periods. This chart illustrates that the only statistically significant difference between the medians of OS in the different studied periods was found at BCLC stage B.

**Table 1 medicina-58-01099-t001:** Division of the cohort according to the key changes in HCC therapy.

Period	1	2	3
Years	2011–2013	2014–2017	2018–2021
Key change	Implementation of BCLC staging as the only staging system at MUH and the start of the systematic application of the HCC surveillance	Standardization of the TACE protocol and start of DAA therapy	Expansion of a systemic oncological therapy

**Table 2 medicina-58-01099-t002:** Characteristics of the cohort.

Characteristic	Number (%) or Mean
Number of patients	170 (100)
Men	136 (80)
Women	34 (20)
Age (years)	69.3 (SD = 8.1)
BCLC	
0 + A	61 (36)
B	53 (31)
C	37 (22)
D	19 (11)
Etiology of cirrhosis	
NASH	67 (39)
ALD	62 (36)
HCV	30 (18)
Cryptogenic	5 (3)
HBV	4 (2)
Other	2 (2)

**Table 3 medicina-58-01099-t003:** Overall survival rates in 1, 3, and 5 years in the 1st period.

BCLC Stage and Period	1-Year OS (%)	3-Year OS (%)	5-Year OS (%)
0 + A1	100	75	62
B1	96	57	46
C1	96	61	N/A
D1	13	0	0

**Table 4 medicina-58-01099-t004:** Overall survival rates in 1, 3, and 5 years in the 2nd period.

BCLC Stage and Period	1-Year OS (%)	3-Year OS (%)	5-Year OS (%)
0 + A2	96	57	46
B2	81	19	5
C2	22	0	0
D2	0	0	0

**Table 5 medicina-58-01099-t005:** Overall survival rates in 1, 3, and 5 years in the 3rd period.

BCLC Stage and Period	1-Year OS (%)	3-Year OS (%)	5-Year OS (%)
0 + A3	96	61	N/A
B3	86	24	N/A
C3	33	11	N/A
D3	0	0	0

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
