# Peer review of "Etiopathogenetic Factors of Hepatocellular Carcinoma, Overall Survival, and Their Evolution over Time—Czech Tertiary Center Overview"

_medicina, 2022, doi:10.3390/medicina58081099_

Round 1
Reviewer 1 Report
This study analyzed HCC patients in the field of liver cirrhosis treated and the effects of key changes in diagnostic and therapeutic procedures in the last 10 years on overall survival (OS) and earlier diagnosis. However, this study is not novatly and lacked the consent of Institutional Review Board. In addition, I had three surggestions:
1. The conclusion need to involve in abstract.
2. In Figure 2&3, the figure legend described unclear, the authors need to explain the linear and compared other factor (NASH, HBV.....et al)
3. In Figure 5, the authors need to show the data (picture) about stage A, C and D.
Reviewer 2 Report
The manuscript entitled “Etiopathogenetic Factors of Hepatocellular Carcinoma, Overall Survival, and Their Evolution over Time – Czech Tertiary Center Overview” reports the changes in the etiology of liver cirrhosis leading to the development of HCC in a tertiary center, during an observation interval of 10 years. Furthermore, the authors tried to present the changes in the stage of HCC at the moment of diagnosis over this time period, as well as the improvements in OS for patients with BCLC stage B HCC (mainly due to the standardization of the TACE).
Major issues
The follow-up of patients with liver cirrhosis aims the detection of HCC as soon as possible. Surprisingly, over a 10-year period, no one patient was diagnosed as stage 0 (very-early stage in BCLC staging). The authors should comment on this observation, as long as based on BCLC scheme of treatment allocation, liver resection is limited to these patients.
The significantly higher rate of BCLC stage A patients diagnosed during the second and the third period vs. the first period could be explained only by the three factors reported by the authors? During the first period was used the screening program for detection of HCC in all the patients with liver cirrhosis from their institution? Maybe during the first period was implemented the screening program in the institution? What exactly means “Implementation of BCLC staging” in the first period?
The treatment of patients in each BCLC stage should be reported for each period (e.g. among the patients with BCLC stage A treated in the first period, xx% underwent liver resection, yy% liver transplantation and zz% ablative therapy; etc.). All the patients with BCLC stage B were treated by TACE? During the first period, BCLC stage B patients received TACE? What type of systemic treatment received the patients diagnosed as BCLC Stage C? Did they receive the same type of treatment during the three periods (e.g. Sorafenib) or the regimens were different (e.g. Atezolizumab plus Bevacizumab, or Sorafenib, or Doxorubicine?). What exactly means “the expansion of the systemic therapy for the advanced stage of HCC” (row 259)?
Was statistically significant the difference in the number/percentage of different BCLC stages during the three periods (it is not mentioned in the Results chapter)? I recommend to consult a statistician to answer this question (although something is mentioned in the Discussion chapter). I also recommend to consult a statistician about the increase/decrease of different etiologies during the 3 study periods (why was evaluated the absolute number of patients when assessed the incidence of ALD, but the percentage of patients in the evaluation of HCV?).
Did the authors received a letter of acceptance from their Institutional Review Board (Ethics Committee) to perform this study?
Minor issues:
There is no Conclusion chapter in Abstract.
Row 51: “When diagnosing a suspected lesion in a risk patient, it is necessary to determine the stage of the disease”. In fact, first of all, it is necessary to establish the certain diagnosis of HCC and subsequently the stage of the disease.
Rows 51-53: “The most used system for determining HCC progression is, with respect to the almost regular current presence of liver cirrhosis, specific” – I do not understand the meaning of this sentence.
Rows 73-74: “The distribution was arbitrary with respect to the changes in the therapy of HCC that the authors considered crucial.”. I consider that the sentence should be reformulated, as the inclusion in one of the three groups was not arbitrary; in fact, division in the 3 periods took into account the changes in treatment allocation strategy and the advent of standardized and/or new therapeutic approaches.
Continuous variables (e.g. age) should be reported as mean +/- standard deviation in case of normal data distribution, or as median [interquartile range 25%-75%] when data are not normally distributed.
I recommend the authors to provide a chart with survival curves for each BCLC stage. It could be possible to group these charts in single figure mentioning each chart with a letter (e.g. Fig. 5a. Comparative OS for BCLC stage A patients; Fig. 5b. Comparative OS for BCLC stage B patients; etc.). The number of patients at risk should be reported in a table placed under each chart. I also recommend to provide the OS rates at 1-, 3- and 5-years (in Results chapter), not only median survival.
Rows 225-226: “Patients with liver cirrhosis with SVR are at a significantly lower risk (by up to 75%) of developing HCC.”. Despite the initial expectations, some authors are reserved to conclude that a SVR is associated with decreased incidence of HCC (it seems that SVR is associated only with decreased risk of progression of HCV-related chronic hepatitis to HCV-related liver cirrhosis).
The main findings of the study (change in the frequency of HCV-related liver cirrhosis during the study period, increased rate of BCLC stage A at the time of diagnosis during the last two periods, as well as the improved OS rates in BCLC stage B patients without a significant change in OS rates for the other stages) should be presented in a more detailed manner during the Discussion chapter.
English language revision is needed.
Round 2
Reviewer 2 Report
The term Table 4 is used for 2 different tables. Please change the second to Table 5.
I recommend to report the survival rates in the Tables 3, 4 and 5 as percentages.